Random forest algorithms for recognizing daily life activities using plantar pressure information: a smart-shoe study

Ren Dian 1 2
Aubert-Kato Nathanael 3 4
Anzai Emi 5
Ohta Yuji 1
http://orcid.org/0000-0002-5669-5337 Tripette Julien 1 2 4 6 tripette.julien@ocha.ac.jp
1 Department of Human and Environmental Sciences, Ochanomizu University , Tokyo , Japan
2 Leading Graduate School Promotion Center, Ochanomizu University , Tokyo , Japan
3 Department of Computer Science, Ochanomizu University , Tokyo , Japan
4 Center for Interdisciplinary AI and Data Science, Ochanomizu University , Tokyo , Japan
5 Department of Human Life and Environment, Nara Women’s University , Nara , Japan
6 Department of Physical Activity Research, National Institutes of Biomedical Innovation, Health and Nutrition , Tokyo , Japan
Keogh Justin
Electronic publication date: 2020 Oct 28
Publication date: 2020
Volume: 8
Electronic Location ID: e10170
Received 2020 May 29; Accepted 2020 Sep 22
Copyright: © 2020 Ren et al.
Copyright year: 2020
Copyright holder: Ren et al.
License: This is an open access article distributed under the terms of the Creative Commons Attribution License, which permits unrestricted use, distribution, reproduction and adaptation in any medium and for any purpose provided that it is properly attributed. For attribution, the original author(s), title, publication source (PeerJ) and either DOI or URL of the article must be cited.
License URL: https://creativecommons.org/licenses/by/4.0/

Keywords: Smart shoes, Activity tracker, Sensor, Activity recognition, Physical behavior, Random forest, Plantar pressure, Physical activity, Health promotion, Wearable

Funding: The authors received no funding for this work.

==============================
Background

Wearable activity trackers are regarded as a new opportunity to deliver health promotion interventions. Indeed, while the prediction of active behaviors is currently primarily relying on the processing of accelerometer sensor data, the emergence of smart clothes with multi-sensing capacities is offering new possibilities. Algorithms able to process data from a variety of smart devices and classify daily life activities could therefore be of particular importance to achieve a more accurate evaluation of physical behaviors. This study aims to (1) develop an activity recognition algorithm based on the processing of plantar pressure information provided by a smart-shoe prototype and (2) to determine the optimal hardware and software configurations.

Method

Seventeen subjects wore a pair of smart-shoe prototypes composed of plantar pressure measurement insoles, and they performed the following nine activities: sitting, standing, walking on a flat surface, walking upstairs, walking downstairs, walking up a slope, running, cycling, and completing office work. The insole featured seven pressure sensors. For each activity, at least four minutes of plantar pressure data were collected. The plantar pressure data were cut in overlapping windows of different lengths and 167 features were extracted for each window. Data were split into training and test samples using a subject-wise assignment method. A random forest model was trained to recognize activity. The resulting activity recognition algorithms were evaluated on the test sample. A multi hold-out procedure allowed repeating the operation with 5 different assignments. The analytic conditions were modulated to test (1) different window lengths (1–60 seconds), (2) some selected sensor configurations and (3) different numbers of data features.

Results

A window length of 20 s was found to be optimum and therefore used for the rest of the analysis. Using all the sensors and all 167 features, the smart shoes predicted the activities with an average success of 89%. “Running” demonstrated the highest sensitivity (100%). “Walking up a slope” was linked with the lowest performance (63%), with the majority of the false negatives being “walking on a flat surface” and “walking upstairs.” Some 2- and 3-sensor configurations were linked with an average success rate of 87%. Reducing the number of features down to 20 does not alter significantly the performance of the algorithm.

Conclusion

High-performance human behavior recognition using plantar pressure data only is possible. In the future, smart-shoe devices could contribute to the evaluation of daily physical activities. Minimalist configurations integrating only a small number of sensors and computing a reduced number of selected features could maintain a satisfying performance. Future experiments must include a more heterogeneous population.

Introduction

The promotion of an active lifestyle among populations remains an on-going problem (Barreto, 2013). Fortunately, the recent boom in the marketing of activity trackers provides new tools to address this issue. The term “activity tracker” is defined as a category of wearable devices, which aims to provide users with feedback on their physical behaviors, physical fitness, and physical activity. This feedback can be provided through a wide variety of parameters, including “step-count,” time spent in activities of selected intensities (sedentary, light, moderate, or vigorous activities), number of floors climbed, and daily energy expenditures (expressed in kilocalories). This type of device has been demonstrated to be effective in supporting active lifestyles and is now widely considered in the development of health promotion policies (Bravata et al., 2007; Bonomi & Westerterp, 2012; Gal et al., 2018; Jennings et al., 2017).

From a technological perspective, the majority of contemporary activity trackers integrate one MEMS 3-axis accelerometer chip that allows the sensing of the user’s body motion. They are typically worn at the hip or wrist and provide feedback on the amount of daily physical activity (Romanzini, Petroski & Reichert, 2012; Kamada et al., 2016). State-of-the-art algorithms directed at evaluating physical behaviors typically feature an activity classification method (Staudenmayer et al., 2009; Ohkawara et al., 2011; Bassett, Rowlands & Trost, 2012). The following are examples of activity classes that are frequently proposed when using the information of one single accelerometer: locomotive vs. non-locomotive vs. mixed activity and sedentary vs. light intensity vs. moderate intensity vs. vigorous intensity activities (Karabulut, Crouter & Bassett, 2005; Oshima et al., 2010). However, latest trackers now feature multi-sensing technologies (e.g., gyroscope, altimeter, light reflectance, thermal resistor), increasing the amount of available information (e.g., inclination, altitude, heart rate, skin temperature) for physical behavior evaluation, and calling for the development of algorithmic suites able to handle the full wealth of available information (Chen & Bassett, 2005; Park et al., 2011).

This trend toward multi-sensing evaluation is expected to benefit from current innovations in the field of wearable technologies and smart clothes, which are designed to work in an interconnected network of 5G devices. In such a fast-evolving context, the current methods could rapidly become outdated, and smart clothes able to collect physiological or mechanical information could assume an ever more central role in the evaluation of physical activity (Intille et al., 2012; Chen et al., 2012, 2016). In the near future, higher-level algorithms will function in the cloud and be capable of collecting available data from a large number of connected devices, and they can select the most relevant information depending on the context to proceed to a continuous and ever more accurate evaluation on physical behaviors. Among these, smart shoes or smart insoles oriented toward the assessment of physical behaviors could be used to evaluate the interaction with the ground and to help refine activity classification.

Medical insoles capable of measuring plantar pressures have already been commercialized as transportable alternatives to force platforms (Fscan; Tekscan, Inc., Boston, MA, USA; ParoTec™; Paromed GmbH & Co. KG, Neubeuern, Germany; PedoSmart). These devices provide a reliable analysis of the center of pressure to assess posture, gait stability, mobility disorders, fall risk, and some other physical considerations. Recently, smart-shoe systems intended for athletes have also been proposed (Nike + Sensor; Nike, Inc., Beaverton, OR, USA; SportProfiler; Digitsol, Nancy, France; Torin IQ; Altra Running, Logan, UT, USA; Mijia; Xiaomi, Beijing, China). They typically provide feedback on plantar pressure distribution, foot landing type, cadence, and contact duration with the ground, among other measurements. To date, smart-shoe systems aimed at monitoring physical behaviors in daily life have only been presented in the scientific literature (De Pinho André, Diniz & Fuks, 2017; Ngueleu et al., 2019). Devices mentioning a high rate of activity recognition typically have multi-sensing abilities, including accelerometer sensors, gyroscopes, temperature sensors, and GPS antennas, providing a large amount of information to the prediction algorithm. However, the inclusion of several in-shoe sensors would likely induce higher production costs as well as challenges for product designers. Furthermore, the high rates of behavior recognition presented in the literature are at times inherent to the study protocols, which may only include a limited number of activities or focus on specific clinical populations, thus preventing the generalization of the results (De Pinho André, Diniz & Fuks, 2017; Ngueleu et al., 2019).

Hence, a smart-insole or smart-shoe system that only uses plantar pressure information and that could recognize multiple human daily life activities has yet to be developed. The present research aims to develop efficient and effective activity recognition algorithms for smart-insole devices featuring 1–7 plantar pressure sensors. Nine daily life activities are considered. The smart-insole prototype used in the present study is equipped with the 7-sensor plantar pressure measurement insole, described elsewhere (Saito et al., 2011; Nakajima et al., 2014). The data analysis is conducted using machine learning methods. The identification of the best hardware and software configurations is conducted through a data processing logical frame, which may be re-used by designers willing to develop smart-shoes devices.

Materials and Methods

7-sensor plantar pressure measurement insole

The shoe hygienic insoles, which are 2 mm thick, were equipped with seven force-sensing resistors (FSR400; Interlink Electronics, Inc., Camarillo, CA, USA). The sensors respond to stimulation ranging from 0.2 to 20 N (8.13–813 kPa), allowing the measurement of human peak plantar pressure (Nandikolla et al., 2017).

The sensors were placed on the heel, lateral midfoot, center of the midfoot, lateral forefoot, center of the forefoot, medial forefoot, and big toe (Fig. 1). They were connected to a 12-bit resolution data acquisition unit with a wireless data transmission sampling rate capacity of 100 Hz, allowing real-time recording during normal ambulatory activities. Insoles with a similar configuration have proven to be valid for the evaluation of posture and gait in previous studies (Saito et al., 2011; Nakajima et al., 2014; Anzai et al., 2020). Multiple pairs of the insole in different sizes were available.

Figure 1 Overview of the smart-shoe prototype and the output.

(A) Pressure sensor location. (B) External view of prototype during experiment with data acquisition system and Bluetooth data transmission unit attached on outside of shoes. (C) Example of raw data time series for the left and right feet at a window of 30 s (activity: walking on a flat surface).

Data collection

The experimental protocol was approved by the Ochanomizu University research ethics committee (#2018-01). A total of 17 female subjects signed written consents and participated in the trial (age: 26 ± 9 years old, weight: 49 ± 3 kg). All the participants were healthy and did not present mobility disorders. The 7-sensor plantar pressure measurement insoles were inserted in a pair of commercial sneakers (Vans Fable 2; VF Corporation, Denver, CO, USA) with stiff and flat midsoles. The insoles and shoes were available from size 22 cm to 27 cm. The participants wore shoes and insoles that best matched their foot size. They performed the following nine activities: sitting, standing, walking on a flat surface, walking upstairs, walking downstairs, walking up a slope, running, cycling, and office work (Table 1). The duration of each activity was approximately 4 min, except “walking on a flat surface” and “running,” which was approximately 8 min. The order in which the 9 activities were completed was randomly selected for each subject. Eleven subjects completed the nine activities. During the course of the experiment, certain subjects expressed a desire to shorten their participation mainly owing to upcoming agenda conflicts, discomfort, or tiredness. Two subjects completed eight activities, two subjects completed seven activities, one subject completed six activities, and one subject completed five activities. For each subject, data for “walking on a flat surface”, “walking upstairs”, “walking downstairs” and “running”, respectively, may have been stored in two files. The final dataset consisted of 196 files corresponding to the 140 activities completed by the 17 subjects. Each file contained 14 independent plantar pressure time series (seven sensors for each of the left and right feet).

Table 1 Description of activities completed by participants of study.

Activity	Description	Type	
Sitting	Chatting and browsing the internet with a smartphone while sitting on an office chair (height: 45 cm, all subjects able to touch the ground with their feet when sitting), indoors	Sedentary	
Standing	Chatting and browsing the internet with a smartphone while standing, indoors	Sedentary	
Walking on a flat surface	Walking on a flat hallway in a campus building, indoors; subjects self-arranged what they considered to be slow and moderate pace (4 min each)	Locomotive	
Walking upstairs	Climbing stairs, indoors; subjects typically completed between 6 and 14 floors	Locomotive	
Walking downstairs	Going downstairs, indoors; subjects typically completed 14 floors	Locomotive	
Walking up a slope	Walking on a treadmill set at a slope of 10%, indoors	Locomotive	
Running	Running on a treadmill at slow and moderate pace (4 min each), indoors; subjects self-selected what they considered to be slow and moderate pace	Locomotive	
Cycling	Riding a utilitarian bike (“mamachari-type”) around the university campus (included turns and changes in pace) at self-selected pace, outdoors	Locomotive	
Office work	Completing several small tasks in a space of approximately 10 m2, including writing and erasing notes on a white table board, carrying light stationeries or files from one desk to another, cleaning up a desk, opening drawers, indoors; all these tasks involved small amplitude movements and displacements only	Mixed (sedentary and locomotive)	

Data preprocessing

The raw data were converted into Newtons (N) and smoothed using a second-order low-pass Butterworth filter with a cutoff frequency of 5 Hz. For each data file, the time series were cut in windows of 1, 5, 10, 15, 20, 25, 30, 35, 40, 45, 50, 55, and 60 s, with an overlap of 50%.

Feature extraction

For each window, 167 data features were extracted using the information from seven sensors on each foot. The features were extracted from the different types of analysis presented in Table 2.

Table 2 Summary of extracted data features.

	Type of analysis	Inputs	Number of features extracted	
General statistics	Average, maximum, SD, median	Each of the 14 sensors	56	
Peak analysis	Peak number, interval between peaks (average, SD), peak magnitude (average, SD), peak width (average, SD)	Each of the 14 sensors	98	
Gait phase	Difference between foot landing and foot lift forces	The envelope of the seven sensors of each foot	2	
Double float duration	
Frequency domain	Power density, mean frequency from 1.67–10 Hz, skewness below 10 Hz, AC component from 2–10 Hz (mean and SD)	Sum of all sensors	5	
Pressure distribution	Difference between forces on the forefoot and heal (mean of the two feet) and correlation (for each foot)	Heel sensor (#1) vs. envelope of sensors located on the forefoot (#4, #5, #6, #7)	6	
Difference between forces on the medial and lateral foot (mean of the two feet) and correlation (for each foot)	Medial forefoot sensor (#6) vs. lateral forefoot sensor (#4)	
Notes:

SD, standard deviation.

The features were grouped into the five following categories:General statistics analysis: The mean, maximum, standard deviation, and median were calculated for each time series. This category included 56 extracted features.

Peak analysis: The peak number, average and standard deviation (SD) of the interval between peaks, average and SD of the peak magnitudes, and average and SD of the peak widths were calculated for each time series using the SciPy library (Jones, Oliphant & Peterson, 2001). The peak widths were calculated at 30% of the peak height. The default parameters of the library were used for the computation of all other features extracted from the peak analysis. This category included 98 extracted features.

Gait phase analysis: The envelope of the signal of the seven sensors was calculated for each foot. For each identified full stance phase, the difference in the force peak yield between the foot contact on the ground (early stance phase) and the foot lift (late stance phase) was calculated and the values averaged over the window. The average duration of the double float phase was also calculated or was set to the null value when such phase does not exist. Two features were extracted in this category.

Frequency domain analysis: The signal of the 14 sensors was summed up and a fast Fourier transform (FFT) was conducted. Preliminary FFT analyses were conducted. The following 5 features were extracted from the AC component of the discrete frequency component series (0.05–50 Hz) and included in the final analysis: (1) power density, (2) frequency signal weighted average from 1.67 to 10 Hz, (3) skewness of the frequency components below 10 Hz, (4) mean of the AC components from 2 to 10 Hz, and (5) standard deviation of the same segment. Events with a frequency lesser than 2 Hz were assumed to be related to the gait cycle. Gait cycle-related behaviors were expected to be described by the features extracted from the above described peak analysis. Moreover, human movements are assumed to not exceed a frequency greater than 10 Hz. Therefore, only the spectral signals at frequencies less than 10 Hz were considered in the present analysis. Five features were extracted in this category.

Pressure distribution analysis: The envelope of the signal of sensors 4, 5, 6, and 7, located in the forefoot area (Fig. 1A), was computed. The difference between the mean of this new series of data and the plantar pressures detected by sensor 1 (heel, Fig. 1A) was calculated for the left and right feet. The difference was averaged to express the anterior–posterior distribution of the plantar pressures. The difference between the mean of the plantar pressures detected by sensor 6 (medial forefoot) and the mean of the plantar pressures detected by sensor 4 (lateral forefoot) was calculated for the left and right feet. The values were averaged to express the medial–lateral distribution of the plantar pressures. Moreover, a Pearson correlation test was used to test the (1) agreement between the envelope of sensor 4, 5, 6, and 7 signals and sensor 1 signal and (2) agreement between the signal of sensor 4 and that of sensor 6. These correlation coefficients were calculated for both the left and right feet. Six features were extracted in this category.

The final number of extracted features depended on the number of sensors included in the processing (cf. paragraphs “Window length,” “Number and location of sensors,” “Number of features”).

Design of activity recognition algorithms

In the present study, the smart-shoe activity prediction algorithms were developed using machine-learning techniques. Data used as input included as many dimensions as the number of features extracted, that is, 167 when using the information from the seven sensors for each foot. Preliminary processing including different machine-learning methods (e.g., k-means clustering, support vector machine) indicated higher performances for the random forest models (results not provided). The analysis presented in this manuscript focuses on the development of random forest models able to process plantar pressure information for activity recognition. The machine-learning analysis was completed using the Python scikit-learn module (Pedregosa et al., 2011).

“Forests” were made of 100 decision trees. Each tree in the forest produced an independent prediction (here, an activity), and the mode of the predictions was chosen as the forest decision. Each tree was constructed using a random subset of the dataset, according to the bagging method described elsewhere (Breiman, 2001). During the construction process, the nodes were successively split until all data points corresponded to the same activity; that is, until the tree’s gini impurity score was equal to zero. This configuration enabled each tree in the forest to output one single prediction (also called a pure decision). Highly informative features could appear in several trees and tended to appear in the nodes that were closer to the root of the trees. Conversely, features with poor discriminating capacities appeared in less nodes across the entire forest (Fig. 2; Supplemental Material 1).

Figure 2 Example of branches for one selected decision tree.

Zoom view on a selected branch of one regression tree of one selected forest. During the training process, the nodes (diamonds) are split until all data points correspond to one activity. At each node, the decision is based on the parameter that best discriminates the sample into two sub-samples. The process is repeated until the generation of a pure offspring, that is, leaves (rounded corner rectangles) containing the data points of one given activity only. The full tree is available in the Supplemental Material 1 (window length: 20 s, configuration: seven sensors, assignment: 1, run: 1).

For the training, the data of six subjects (i.e., approximately 33% of the dataset) were used under five different subject-wise assignments (Fig. 3). At the training stage, only the data of the subjects who completed the nine activities were used. A total of 20 training-test runs were performed for each assignment, with each run using different random subsets of the dataset (hereafter called “random states”). For the testing process, the generated random forest modules evaluated the data of the remaining 11 subjects. The results averaged across all five assignments (i.e., across 100 forests), were presented as confusion matrices of the predictions vs. actual activities. The results were also presented as mean (minimum, maximum) when summarizing the overall performance across all activities. All the results presented in this manuscript correspond to the outcome of the evaluation of the random forest modules using the test samples only. None of the reported scores are related to the training phases.

Figure 3 Overview of the machine-learning procedure.

Training-test ratio: 6–11 (subject-wise). Testing method: multi-hold-out (data assigned to five different training-test combinations). For each assignment, 20 runs are conducted using different random subsets of the data set (or “random states”). Blue: training samples. Salmon: test samples.

Data analysis framework

As illustrated in Fig. 4, the analyses are integrated in a 3-stage logical flow.

The window length analysis aims at identifying the optimum analytic window length.

The analysis of a pre-selected set of 25 sensor configurations, that is, configurations using the information of different numbers of sensors and/or the information of sensors placed at different locations, aims at identifying the best hardware combination for each possible number of sensors ranging from 1 to 6 (the 7-sensor configuration only has 1 possible combination). This analysis was conducted using the optimum window length identified in (1).

A final analysis exploring the contribution of each feature to the forest outputs aims at finding the most efficient number of features to be used for each of the seven best sensor configurations identified in (2). Again, this analysis was conducted using the optimum window length identified in (1).

Figure 4 Chart of the 3-stage data processing flow.

Stage 1: “window length”. Stage 2: “number and location of sensors”. Twenty-five configurations were selected among the 127 possible combinations of sensors (see “Materials and Methods”, “Stage 2: number and location of sensors”). Some of these configurations were expected to perform well (green bars). Some of these configurations were expected to perform poorly (pink bars). Stage 3: “number of features”. The orange/dotted connectors indicate the logical links between each stage of the analysis.

Further details related to each of the three stages are given in the three following subsections.

Stage 1: window length

The described analysis was performed for different window lengths (1, 5, 10, 15, 20, 25, 30, 35, 40, 45, 50, 55, and 60 s) using the data of the seven sensors per shoe with each data point having 167 dimensions corresponding to the maximal number of data features that were possible to extract. The optimum window length was defined at the point where the slope of the function describing the prediction rate vs. window length began to decrease. This optimum window length was used for all subsequent analysis.

Stage 2: number and location of sensors

The processing was repeated from scratch for different sensor configurations, that is, different location and/or number of sensors, for the optimal window length only. Twenty-five configurations were selected among the 127 possible combinations of sensors. The selection was performed using subjective criteria: (1) reproduction of a selection of the configurations found in the literature or in the industrial sector of running shoes, (2) selection of combinations allowing the collection of relevant information for the prediction of gait and postural behaviors, and (3) selection of combinations that are believed to miss some pieces of relevant information for the prediction of gait and postural behaviors (Fig. 4). Seventeen configurations were selected with respect to criteria (1) and (2). These configurations were expected to perform well. Eight configurations were selected with respect to the criterion (3). These configurations were expected to perform poorly. The number of dimensions of the data points decreased in accordance with the decreased number of sensors. For certain configurations with the same number of sensors, the data points present different numbers of dimensions. Indeed, as indicated in Table 2, some features may need specific sensor locations to be computed. All the tested configurations are noted in Fig. 4.

Stage 3: number of features

The processing was again repeated from scratch with the best configurations only and for a decreasing number of features, which were removed one-by-one based on their discriminating capacities (Fig. 4). For each sensor configuration, the analysis was performed with the maximum number of available features (similar to what was performed for the previous process, cf. paragraph “Number and location of sensors”). Features were ranked relative to their discriminating capacities, that is, from the highest to lowest informative feature, across the 100 runs of the analysis (five assignment × 20 random states, cf. “Prediction algorithm: training and test”). The lowest informative feature was removed from the dataset, and a new repetition of training-test runs was performed. The entire process was repeated until only one feature remained. The minimum number of features corresponding to the inflection point for the prediction rate vs. number of the feature was considered to be the optimum number of inputs. A total of 686 combinations of sensor configurations and number of features were tested (i.e., best 1-sensor: 29, best 2-sensor: 54, best 3-sensor: 76, best 4-sensor: 98, best 5-sensor: 120, best 6-sensor: 142, 7-sensor: 167).

Results

The “prediction rates” and “rates of good predictions” presented in the text and figures refer to the accuracy, calculated as follows: correctly predicted sample/total number of samples. When reporting statistical results, the terms “average” and “mean” point to the average accuracy across the 100 forests of one round of evaluations (see Fig. 3). The expressions “best single forest” and “best performer” refer to the one single forest that showed the best accuracy score among the 100 forests produced for one round of evaluations. Conversely, the term “worst single forest” points to the one single forest that showed the worst accuracy score among the 100 forests produced for one round of evaluations. Logical links between the 3 stages of the analysis are shown in Fig. 4. The values indicated at the intersections of “true label” and “prediction” in the confusion matrices refer to sensitivity, calculated as follows: true positives/(true positives + false negatives).

Stage 1: window length

The average performances of the 7-sensor configuration tested at different window lengths (1, 5, 10, 15, 20, 25, 30, 35, 40, 45, 50, 55, and 60 s) are presented in the Fig. 5. The full set of 167 features was used for all tests. The best prediction rate was obtained with a 45-s window length: 0.90 (min: 0.86, max: 0.91). A 20-s length was associated with an average of 0.89 (min: 0.82, max: 0.91). The average prediction rates for window lengths between 20 and 60 s showed marginal variations within the 0.89–0.90 range. To preserve the highest possible temporal resolution for future applications, 20 s was selected as the optimum length. The rest of the analyses were conducted using a 20-s window length.

Figure 5 Window length effect on activity recognition rate.

The tests were completed using the full set of 167 features available for a configuration of seven sensors per shoe and for selected window lengths. Pink boxes: 1, 5, 10, and 15 s. Green box: 20 s (considered optimum). Yellow boxes: 25, 30, 35, 40, 45, 50, 55, and 60 s. Red diamonds: mean values.

As indicated in Fig. 6, “walking up a slope” could be confused with “walking on a flat surface” or “walking upstairs.” Confusions between “walking upstairs” and “walking downstairs” and between “standing” and “office work” were noted to a certain extent depending on the window length.

Figure 6 Confusion matrices for four selected window lengths.

Normalized sensitivities are averaged for each activity across the 100 forests. (A) 1-s window length. (B) 20-s. (C) 30-s. (D) 45-s. The tests were completed using the full set of 167 features available for a configuration of seven sensors per shoe.

Stage 2: number and location of sensors

The average performances of a subset of 25 selected sensor configurations are presented in Fig. 7. For each configuration, the analyses were performed using all the available features. The best average prediction rate was 0.89. In addition to the 7-sensor configuration, this rate was observed for the four following configurations: 6 sensors, 145 (heel, lateral midfoot, lateral forefoot, medial forefoot, center of the midfoot, center of the forefoot): 0.89 (min: 0.82, max: 0.92)

6 sensors, 142, (heel, lateral midfoot, lateral forefoot, big toe, center of the midfoot, center of the forefoot): 0.89 (min: 0.83, max: 0.91)

5 sensors, 120 features (heel, lateral midfoot, lateral forefoot, center of the midfoot, and center of the forefoot): 0.89 (min: 0.85, max: 0.92)

4 sensors, 98 features (heel, lateral midfoot, lateral forefoot, center of the forefoot): 0.89 (min: 0.85, max: 0.92)

Figure 7 Performance of activity recognition of random forest algorithms for 25 sensor configurations.

The number of features depended on the number and location of the sensors. The number of features from left to right, one sensor: 29, 29; two sensors: 54, 54, 54, 51; three sensors: 76, 76, 76, 73; four sensors: 98, 101, 98, 98; five sensors: 120, 123, 123, 123, 120, 120; six sensors: 142, 145, 145, 142; and seven sensors: 167. Green boxes: sensor configurations that were expected to perform well (positions 1, 2, 3, 4, 7, 8, 11, 12, 13, 15, 16, 17, 18, 21, 22, 23, 25 from left to right). Pink boxes: sensor configurations that were expected to perform poorly (positions 5, 6, 9, 10, 14, 19, 20, 24 from left to right) (cf. Fig. 4). Red diamonds: mean values.

Regarding the best performers, selected forests achieved a prediction rate of 0.92. This result was obtained with a 3-sensor configuration only (heel, lateral midfoot, center of the forefoot). All selected configurations with at least two sensors produced an average rate of good predictions of 0.80 or more. All the configurations with at least five sensors produced an average rate of good predictions of 0.85 or more. All the configurations with at least two sensors, which were expected to perform well, produced an average rate of good predictions of 0.87 or more. Certain forests with one sensor located at the center of the forefoot could compute prediction rates as high as 0.86. The mean and maximum rates of good predictions of a larger panel of 67 selected configurations are presented in Supplemental Material 2.

The confusion matrices presented in Figs. 8 and 9 indicate the sensitivity score of each activity, for the best and worst 1-, 2-, 3-, 4-, 5-, and 6-sensor configurations. Among the best sensor configurations, the decrease in the average prediction rate observed when reducing the number of sensors from two to one might be explained mainly by the higher levels of confusion between “sitting” and “standing” and between “walking downstairs” and “walking upstairs.” For example, the best 1-sensor configuration wrongly predicted “walking upstairs” instead of “walking downstairs” in 36% of the cases. Among the worst configurations, the decrease in the average prediction rate observed when reducing the number of sensors from four to three could be explained mainly by a decrease in sensitivity for “cycling” (0.91 and 0.80).

Figure 8 Confusion matrices for the best configuration depending on the number of sensors and using all the available features.

Normalized sensitivities are averaged for each activity across the 100 forests. (A) Best 1-sensor configuration. (B) Best 2-sensor configuration. (C) Best 3-sensor configuration. (D) Best 4-sensor configuration. (E) Best 5-sensor configuration. (F) Best 6-sensor configuration. N: number of features (all available features).

Figure 9 Confusion matrices for the worst configuration depending on the number of sensors and using all the available features.

Normalized sensitivities are averaged for each activity across the 100 forests. (A) Worst 1-sensor configuration. (B) Worst 2-sensor configuration. (C) Worst 3-sensor configuration. (D) Worst 4-sensor configuration. (E) Worst 5-sensor configuration. (F) Worst 6-sensor configuration N: number of features (all available features). The 1-sensor configuration presented in this figure was initially expected to perform well. It corresponds to the worst 1-sensor configuration among the two selected in this study (cf. Fig. 7).

Stage 3: number of features

The changes in performance of the seven selected configurations when decreasing, one-by-one, the number of features used for the prediction are displayed in Fig. 10. For these configurations, the mean rate of good predictions increased from an average 0.46 ± 0.03 when using one feature to 0.87 ± 0.04 when using a set of 20 high performance features. Using 20 features only, all the selected configurations demonstrated a mean rate of good predictions greater than 0.85, with the worst single forest scoring at 0.81 (2-sensor configuration), except for the 1-sensor configuration, which demonstrated a rate of 0.78 (min: 0.72, max: 0.83). The data are presented in Supplemental Material 2. The overall performance remained constant when the predictions were computed with more features. The mean rate of good predictions exhibited an average of 0.87 ± 0.03 when considering computations performed using the maximum number of available features, that is, 29, 54, 76, 98, 120, 142, and 167, respectively, for the selected 1-, 2-, 3-, 4-, 5-, 6-, and 7-sensor configurations.

Figure 10 Effect of the number of features on the activity recognition rate.

(A) Results for all identified best 1-, 2-, 3-, 4-, 5-, and 6-sensor configurations (cf. Fig. 8) and for the 7-sensor configuration. (B) The 7-sensor configuration. (C) Best 6-sensor configuration. (D) Best 5-sensor configuration. (E) Best 4-sensor configuration. (F) Best 3-sensor configuration. (G) Best 2-sensor configuration. (H) Best 1-sensor configuration.

Considering a 20-feature cut-off below which features became increasingly important, 44 important features were identified over the 20–140 alternatives enabled by the seven selected configurations (Fig. 11). Seven features systematically ranked among the 20 most important features of the seven selected configurations: average peak interval of the left foot heel sensor (peak analysis), average peak magnitude of the right foot heel sensor (peak analysis), mean of the AC component (frequency domain), number of peaks for the right foot heel sensor (peak analysis), number of peaks for the left foot heel sensor (peak analysis), standard deviation of the left foot heel sensor plantar pressures (general statistics), and standard deviation of the AC component (frequency domain). Among the 44 important features, 24 belong to the “peak analysis” category, 16 to the “general statistics” category, 3 to the “frequency domain” category, 1 to the “gait phase” category, and 0 to the “pressure distribution” category. Regarding the 7-sensor configuration only, the features related to the heel and central forefoot were identified five times. No feature directly extracted from the analysis of the big toe pressure ranked among the set of important features.

Figure 11 Identification of most important features.

Results are displayed for the previously identified best 1-, 2-, 3-, 4-, 5-, and 6-sensor configurations (cf. Fig. 6) and for the 7-sensor configuration. Red: identified as an important feature in all seven selected configurations. White: identified as an important feature in one selected configuration only. FFT: fast Fourier transform; SD: standard deviation.

The best 1-sensor configuration (heel sensor) using the single most informative feature demonstrated a mean rate of good predictions of 0.43 (min: 0.41, max: 0.44). Only the “running” and “sitting” activities demonstrated a sensitivity score greater than 50% (Fig. 12). As indicated in Fig. 12, the other selected sensor configurations were associated with sensitivity scores of 82% or more for all activities except “office work,” “walking up a slope,” and “walking upstairs,” when using a limited number (i.e., 9–23) of features.Regarding the 7-sensor configuration specifically, the confusions noted when using the 23 most informative features (Fig. 12D) were similar to the ones noted when using the full number of available features (Fig. 6B), except for “walking downstairs” and “walking up a slope,” which had better sensitivities (0.92 vs. 0.87 and 0.68 vs. 0.63, respectively) when using 23 features only. That phenomenon can be explained by the greater difficulty to fit a classifier with a higher number of dimensions. In theory, the same performance should be attainable with more features, at the risk of overfitting the system and decreasing its generality (performance on unknown data) (Lever, Krzywinski & Altman, 2016). Confusion matrices of some selected single forests produced with the best 4-sensor configuration are presented in Fig. 13. A low performance single forest with a relatively high number of features (49 over a maximum of 98 available) was associated with low sensitivity scores for the “office work,” “walking downstairs,” and “walking up the slope” activities (0.74, 0.61, and 0.56, respectively), consistent with the pattern that has been frequently found on confusion matrices, as displayed in Figs. 6, 8, and 9. Interestingly, the worst single forest among the ones built with 29 features only had the highest sensitivity for the “walking up the slope” activity (0.70). Finally, confusion matrices of the best single forests built with 86 and 22 features demonstrated a similar pattern of missed predictions, with “office work” and “walking up the slope” being relatively poorly recognized (<0.85 and <0.65, respectively).

Figure 12 Confusion matrices for four selected configurations using different numbers of features.

Normalized sensitivities are averaged for each activity across the 100 forests. (A) 1-sensor configuration: heel. (B) 3-sensor: heel, lateral midfoot, center of forefoot. (C) 5-sensor: heel, lateral midfoot, center of midfoot, lateral forefoot, center of forefoot. (D) 7-sensor: all. N: number of features.

Figure 13 Confusion matrices for selected forests captured from the best 4-sensor configuration.

Values refers to the normalized sensitivity of the selected forest for each activity. (A) One selected bad performer with a relatively low number of used features. (B) One selected good performer with a relatively low number of used features. (C) One selected bad performer with a relatively high number of used features. (D) One selected good performer with a relatively high number of used features. “Best” refers to the results obtained from the forest with the highest prediction rate. “Worst” refers to the results obtained from the forest with the lowest prediction rate. “Performer” here refers to one single forest. N: number of features.

Supplementary results

A more comprehensive analysis has been conducted considering a larger panel of 67 sensor configurations. Random forest modules were systematically created and tested for each window length candidates (1–60 s) and each possible number of features (maximum to one), without any selection of the best sensor configurations like in the 3-stage data processing flow presented in Fig. 4 (the results of which have been presented in the above three subsections). The machine learning procedure was the same as the one detailed in the method section. Therefore, 75,178 additional analyses have been completed, resulting in the computation of 7,517,800 forests.

These supplementary analyses were associated with higher prediction scores, highlighting the whole potential of using plantar pressure data for the recognition of physical behaviors. As shown in the Supplemental Material 3, 297 sensor configurations were associated with at least one forest presenting a prediction score of 0.92 or more. Regarding the highest scores, at least 12 forests presented a rate of good predictions of 0.94. The best average scores ranged from 0.54 to 0.91, a scale similar to the one of the results of the 3-stage analysis (Supplemental Material 2). Eighty-seven sensor configurations were associated with average rates of good predictions of 0.90 or more. As shown in Supplemental Material 4B, the best performances observed in these analyses are systematically associated with window lengths of 30 s or longer. Regarding the identification of an optimum analytic window length, the results still points to a period of 20 s (Supplemental Material 4A). The question of the time resolution is discussed later in the manuscript. The results of these supplementary analyses are summarized in Supplemental Materials 3 and 4.

Discussion

In the present study, homemade smart shoes mounted with seven pressure sensors were used to collect plantar pressures during nine daily life activities. From the plantar pressure data, 167 features bearing a potential interest for the characterization of gait and posture were extracted. Random forest models using subject-wise training-test assignments were utilized to develop smart-shoe activity recognition algorithms. A 20-s window length was identified as the optimal period for the extraction of the features. Forests could recognize activities at an average rate of good predictions of 0.89, with certain single forests demonstrating a rate as high as 0.92. Reducing the number of sensors to two (heel and lateral forefoot) and selecting 20 high performance features maintained the average rate of good predictions above 0.85.

Performances

Smart shoes in their maximal configuration (i.e., 7 sensors per foot and 167 features extracted from the collected plantar pressures) allow random forest modules to recognize activities at a rate of good predictions of 0.89 (min: 0.82, max: 0.91). Each single activity was associated with a sensitivity score of at least 0.87, except “office work” and “walking up a slope,” which presented lower scores (0.80 and 0.63, respectively) (Fig. 6B). “Office work” was confused with “standing” in 18% of cases. The latter is not surprising considering the content of the “office work” activity, which includes a considerable number of tasks realized in the standing posture. Numerous subjects consumed a significant amount of time writing and erasing notes on a white table board while performing the “office work”-labelled activity. This could have created this confusion with the “standing” activity. Moreover, poor predictions involving the “walking up a slope” activity being confused with “walking upstairs” or “walking on a flat surface” was a recurrent issue of the present analysis. This type of confusion occurred regardless of the sensor configuration or the number of features used as input. Depending on the field of application, several of the above-mentioned confusions could have marginal or significant consequences on the final evaluation of physical behaviors. Future smart-shoe studies should also consider extracting data features that are more likely to report on slope-related gait alterations.

Conversely, random forest module outcomes indicated only a small number of confusions for the “cycling,” “running,” or “sitting” activities. Although “running” and “sitting” are typically well recognized in research protocols that use accelerometer sensors, which remain the current primary hardware choice for activity trackers (Pavey et al., 2017; Trost, Zheng & Wong, 2014; Voicu et al., 2019), the recognition of “sitting” behaviors has actually proven technically challenging in real-life conditions (Kerr et al., 2018). Extrinsic behavioral factors, such as people leaving their tracking device to charge when they are resting or sitting, render the assessment of sedentary behaviors even more difficult. In the present study, smart shoes demonstrated high level of sensitivity for “sitting” (0.96 or more for any of the selected configurations with at least two sensors and a window length of 20 seconds) and low level of confusion with the other sedentary activity (i.e., “standing,” (0.00–0.01), except for some 1- and 2-sensor configurations). Such outcomes should be considered as promising for the monitoring of sedentary behaviors outside the house.

Finally, differences were noted among the single forests for the performance in each activity. For example, one forest tagged with a low overall performance displayed in Fig. 13 performed surprisingly well for the recognition of the “walking up a slope” activity. However, this enhanced performance would appear to be possible at the expense of an altered sensitivity for other activities. This may reflect the capacity of random forest modules to specialize for one given type of activity. Further analyses, which are beyond the scope of the present report, would be necessary to identify the “ins and outs” of forest specialization and determine if the random forest method could be adapted to the specific case of smart shoes to obtain more homogenous recognition rates across activities. For example, forests with a higher number of trees or hierarchical models assigning data points to sub-classes before proceeding to the final evaluation could be considered for future studies.

Comparison with previous studies and originality

Several reviews have summarized the outcomes of studies interested in the validity of instrumented insoles developed for activity recognition (De Pinho André, Diniz & Fuks, 2017; Ngueleu et al., 2019). Similar to the observations in the present research, specific studies have reported excellent performances, with rates of good predictions scoring frequently over 0.90. However, they can also be linked with experimental limitations, altering the generalization of the results, such as a small number of tested activities, small number of subjects, special groups of individuals, and training-test procedures completed separately for each subject. Moreover, the majority of these studies have used hardware with multi-sensing capabilities. Hegde et al. (2017) developed the SmartStep system, which featured three pressure sensors, one 3-axis accelerometer, and a gyroscope. The pressure sensors were placed at the heel, first metatarsal head (i.e., equivalent to the medial forefoot), and big toe. They tested the activity recognition capabilities of the SmartStep system for a wide range of daily life activities. Similar to the present report, they observed an average rate of good prediction of approximatively 0.90. They also reported recurrent mis-predictions for “walking downstairs” (0.62), which is frequently confused with “walking on a flat surface” and “walking upstairs” and for “shelving items” (0.61), the description of which resembles the “office work” activity of the present study, and which is frequently confused with “standing.” Smart shoe-based activity recognition projects appear to be associated with redundant challenges related to ascending and/or descending locomotive activities and activities combining locomotive and non-locomotive behaviors. Moreover, in another recent study, Moufawad el Achkar et al. (2016) used a simple decision tree classifier to achieve excellent rates of good predictions for nine activities, including “walking downstairs” (0.98), “walking upstairs” (0.99), and “walking uphill” (0.96). However, their smart-insole system featured a barometer in addition to eight pressure sensors, one 3-axis accelerometer, one 3-axis gyroscope, and one 3-axis magnetometer, which surely helped the assessment of ascending and/or descending locomotive behaviors.

According to Ngueleu et al. (2019), smart shoe-based activity recognition studies that only use plantar pressure information are limited. Although some of these studies reported acceptable performance, protocols were typically limited to a small number of locomotive behaviors (Zhang et al., 2005; Zhang & Poslad, 2014), small number of subjects, or training-test procedures completed separately for each individual (Sugimoto et al., 2010). Therefore, the present research provides important findings to the relatively small corpus of knowledge on plantar pressure-based activity recognition. Other originalities of the present research include the use of a random forest modeling method to develop different activity classifiers and a comparison of different sensor configurations (number and location) within one single experimental protocol.

Temporal resolution, sensor configuration, number of features, manufacturing, and algorithmic considerations

Although windows of 30 and 45 s were linked with better performances for the recognition of “office work” and “walking up a slope”, overall, the performances were consistent across all analyses performed with a window size of 20 s or longer (Figs. 6B–6D). In real-life situations, a short window length reduces the probability of overlapping activities over the span of one analytic period. Therefore, a 20-s length with a 50% overlap between windows was selected as the optimum window length. It allowed computing predictions every 10 s. Considering future applications, this relatively high temporal resolution would allow applying a second statistical algorithmic layer consisting of comparing the prediction of one given window with the ones of its neighbors (Witowski et al., 2014). This would provide the opportunity to have a set of six “instant” predictions to determine the dominant behavior every minute. Further explorations that include free-living experiments are necessary to elaborate further on the issue of temporal resolution.

One interesting finding of the present study is the marginal alteration of the overall performance obtained with a reduced number of sensors. Although configurations without the heel sensor systematically present lower performances, other configurations that include at least two sensors demonstrate average rates of good predictions of 0.87 or more (Fig. 7). The absence of a heel sensor appears to worsen confusions between ascending and descending activities and between “office work” and “standing” (Fig. 8 and 9). Using one sensor only, the average rates of good predictions declined below 0.80. Furthermore, marginal variations of the overall performance were noted when reducing the number of features down to approximately 20 (Fig. 10). The reduction of the number of features given to the forests was accomplished in a manner that favored the most contributive features. Extracts from the FFT and peak analyses were redundant in the lists of 20 important features (Fig. 11). However, this result could also be the mere reflection of the higher number of gait activities included in the present protocol, which all present cyclic plantar pressure patterns. Therefore, future studies should include a more balanced number of locomotive, non-locomotive, and mixed activities to determine whether this trend is confirmed or not. Moreover, no feature extracted directly from the big toe sensor ever scored among the 20 most important features. This location may not be relevant for smart-shoe prototypes aimed at behavior recognition. Although the 167 data features were selected to be as comprehensive as possible and to accommodate the analysis on the reduction of the number of sensors, the list of potentially informative features is not closed. Future studies could propose extracting different features to boost the performance on a similar or different subset of activities. With respect to the above-discussed results, shoe manufacturers willing to develop activity recognition devices should probably consider the opportunity to implement a minimalist sensor configuration instead of the full 7-sensor configuration. They should also consider the relevance of using an exhaustive number of features, whereas a subset of 20 features has been demonstrated to perform equally well. All these considerations will influence shoe design (relative to the location of sensors and other hardware), microprocessor selection (relative to the computational needs), and, ultimately, the financial cost of the device (Eskofier et al., 2017).

Limitations and strengths

Some characteristics of the present protocol could limit the interpretation of the results presented in this report and should be mentioned clearly for the readers. First, the current protocol only includes nine different daily life activities. This number puts the study among smart-shoe protocols testing a large sample of activities (Ngueleu et al., 2019). The challenges related to the recognition of activities that potentially present closed plantar pressure patterns are addressed in an adequate manner. However, a larger number of activities should be studied in the future to reflect more exhaustively physical behaviors of the daily life, for example, sport activities and a wider panel of activities combining locomotive and non-locomotive behaviors. Second, the experimental design does not include further validation of forest performances in real-life situations. Similarly, no comparison with commercial activity monitors was performed. Future protocols should include a free-living validation to increase the generalization of the results to real-life situations. Third, the present protocol includes a relatively homogenous population. Subjects were all healthy women. To address this potential issue, five different subject-wise training–test assignments were used to develop and test the forests. In addition, a conservative 6–11 training-test assignment ratio has been used to limit the wealth of the available information during the training phase and create more challenging conditions relying on inter-individual differences. However, a more heterogeneous sample of the population must be tested before generalizing further the results of the present study. A more heterogeneous population would indeed provide a more diverse information to the training algorithms, which could also result in increased good prediction scores. Overall, given the homogeneity of the population used in the present study, one should exercise caution when interpreting the results. The best configurations identified in the present study could differ from one population to the other. Designers are therefore encouraged to select a subject sample large enough to be representative of the targeted population and provide the wealthiest possible information to the machine learning algorithms. Finally, the present work does not address the question of a multi-sensing environment. Given that alternative sensing options could already be embedded in other type of devices (e.g., activity trackers, smartphones), one could consider that smart shoes should primarily specialize in the collection of information on the foot–ground interaction. The present protocol allows focusing on the sole performance of plantar pressure-based activity recognition to assess the relevance of including smart shoes in a network of devices dedicated to physical activity evaluation (Chen et al., 2016; Eskofier et al., 2017).

Conclusions

In this work, random forest modules as behavior recognition algorithms for plantar pressure measurement using smart shoes were explored and proved relevant. Indeed, smart shoes mounted with seven pressure sensors and extracting 167 plantar pressure data features could recognize nine different daily life activities with an average of good prediction of 0.89. Interestingly, the results suggest a marginal reduction of performance for configurations downgraded to two, three, four, five, or six sensors and the computation of approximately 20 plantar pressure data features, which could ease the design and manufacturing of smart-shoe products. Future studies are necessary to generalize the present findings to a larger sample of the population and larger number of behaviors. Considering the trend toward the development of wearable devices with 5G capacities, smart shoes could become a crucial element of systems allowing self-monitoring of physical activity, thus having an important role in promoting active and healthy lifestyles.

Supplemental Information

Supplemental Information 1 One example of one decision tree in one selected forest.

The tree is composed of 101 nodes and leaves. This number may vary from trees to tress. During the training process, nodes are split until all data points correspond to one activity. At each node, the decision is based on the parameter that best discriminates the sample in two sub-samples. The process is repeated until the generation of pure offspring, i.e. leaves containing data points for one given activity only. Gini: sample impurity, score from 0 to 1, with 0 indicating pure offspring. Samples: number of data point evaluated by the nodes. Value ([cycling, downstairs, office, run, sitting, slope, standing, upstairs, walking]): weight of each activity in the evaluated sample; 0 indicates the absence of data point for one given activity. Class: activity with the most data point. The tree is extracted from the following forest: window length: 20 sec, configuration: 7 sensors, assignment: 1, run: 1.

Click here for additional data file.

Supplemental Information 2 Summary of results for a larger panel of 67 sensor configurations (window length: 20 seconds, numbers of features: maximum, 20, best).

Click here for additional data file.

Supplemental Information 3 Summary of results for a larger panel of 67 sensor configurations (window lengths: all, number of features: best).

Blue: best single forest. Orange: best average rate of good predictions.

Click here for additional data file.

Supplemental Information 4 Summary of the supplementary analyses following the formats of Figure 5 and 7.

(A) window length effect on activity recognition rate (best combination for “number and location of sensors” and “number of features”). Pink boxes: 1, 5, 10 and 15 seconds. Green box: 20 seconds (considered optimum). Yellow boxes: 25, 30, 35, 40, 45, 50, 55 and 60 seconds. Red diamonds: mean values. (B) Performance of activity recognition of random forest algorithms for 25 sensor configurations (best combination for “window length” and “number of features”). Green boxes: sensor configurations that were expected to perform well. Pink boxes: sensor configurations that were expected to perform poorly. Red diamonds: mean values. (C) Performance of activity recognition of random forest algorithms for 25 sensor configurations (“window length”: 20 seconds, “number of features”: best average rate of good prediction). Green and pink boxes: same chart as for panel B.

Click here for additional data file.

The authors thank the subjects who participated in the study. The authors also thank Dr. Natsuko Nagasawa and Meina Wang for their support and assistance with this project.

Additional Information and Declarations

Competing Interests

Author Contributions

Human Ethics

Data Availability

The authors declare that they have no competing interests.

Dian Ren conceived and designed the experiments, performed the experiments, analyzed the data, prepared figures and/or tables, authored or reviewed drafts of the paper, and approved the final draft.

Nathanael Aubert-Kato performed the experiments, analyzed the data, prepared figures and/or tables, authored or reviewed drafts of the paper, and approved the final draft.

Emi Anzai performed the experiments, prepared figures and/or tables, and approved the final draft.

Yuji Ohta performed the experiments, authored or reviewed drafts of the paper, and approved the final draft.

Julien Tripette conceived and designed the experiments, performed the experiments, analyzed the data, prepared figures and/or tables, authored or reviewed drafts of the paper, and approved the final draft.

The following information was supplied relating to ethical approvals (i.e., approving body and any reference numbers):

The experimental protocol was approved by the Ochanomizu University Research Ethics Committee (#2018-01).

The following information was supplied regarding data availability:

The code is available in GitHub: https://github.com/dian-R/CodeForSmartShoesPeerJ.

The data are available at Zenodo: Ren, Dian, Aubert-Kato, Nathanael, Anzai, Emi, Ohta, Yuji, & Tripette, Julien. (2020). Data for: “Random forest algorithms for recognizing daily life activities using plantar pressure information: A smart-shoe study” [Data set]. PeerJ. Zenodo. DOI 10.5281/zenodo.4050390.

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
