# Peer review of "Random forest algorithms for recognizing daily life activities using plantar pressure information: a smart-shoe study"

_PeerJ, doi:10.7717/peerj.10170_

## Round 0.1 · original submission · Major Revisions

The two reviewers and I are impressed with many aspects of the research design and manuscript. Please look to address all the comments of the two reviewers before resubmitting this manuscript for peer review.

Reviewer 1 ·

Basic reporting

no comments

Experimental design

no comments

Validity of the findings

no comments

Additional comments

The paper is well structured and has systematic exploration of using plantar pressure for activity classification. However, due to excessive details about sensor and feature selection, the paper is a little difficult to understand specifically for results. Here are some comments:

1) I generally recommend the authors to remove the descriptions about energy expenditure in the paper, considering the paper does not work on algorithm related to energy expenditure and the paper focuses on activity classification only.

2) I recommend the authors to rewrite result section. The authors used all sensors and features configuration to determine optimal window size. And then use the determined window size for the following sections. This raised a question that maybe in other configuration of sensor and features, this window size is not optimal. Please give more explanations about this. Why do the authors simplify the selection of window sizes while selecting features and sensors?

3) In abstract, please give a concise description for validation method? Like cross validation or hold-out validation?

4) In Line 40-43, all prediction accuracy percentage is obtained under window size of 20 seconds?

5) For one category of activity, sitting, there could be some confounding factors which influence the measurements? Could the author release information about height of the chair? As we know, if the subject's lower leg is shorted than the height, the foot will be not on the ground while sitting and plantar pressure is 0. In some cases, the foot could be on the ground.

6) In line 74-75, nature of acceleration data does not allow refining of the classes until all daily life activities are identified. It is unclear and why?

7) In 232-233, the authors mentioned 20 train-validation runs? so all runs is on training dataset (6 subjects)? Is it cross validation? If the reviewer's understanding correct, the validation dataset (11 subjects) might be better called testing dataset to avoid confusion?

8) From 240-274, the window size, number of sensors and features are all determined by training dataset? Please ensure not to touch the validation dataset while doing sensor and feature selection and hyper-parameters configuration. The dataset deriving reported accuracy percentage should not be used in model configuration.

9) From line 282-284, the authors first mention the best accuracy is achieved at 45s, and then what does best forest mean at 30 seconds? And then why the author select 20 seconds as the optimal window size. Please give clear explanations of rules about selection of window size.

10) In number of sensors and features subsection, the protocol of evaluation is not clear. Did the authors first determined sensor number while using all features? And then fixing the optimal sensor number and then work on feature selection? Is this understanding correct? It seems there are a lot of combinations of sensors and features? For 7 sensors, the total number of combinations is 7*6*5....*1. If considering the feature number, the combination is more various. The reviewer accept some degree of simplification, but please give more clear explanations.

Reviewer 2 ·

Basic reporting

The rational for this study is clearly presented, with appropriate links to literature in the field. Authors provided sufficient field background, relying on adequate references.

Experimental design

Authors used meaningful methods, which are well described. I only have two questions:

1. Did participants perform the 9 tasks in a pre-defined order or randomly-selected order? This can possibly influence the performance of algorithms to recognize activities.

2. To train algorithms, authors used a five-different subject-independent training–validation assignments. Would a higher number of subject for training have changed the performance of algorithms regarding good predictions?

Validity of the findings

Results are presented with details; text, figures and tables complement each other. Even the algorithms showed acceptable performance, there are room for improvement. This inspire my following question:

The average good prediction was 0.89, which is good but not perfect. Would a more heterogenous sample have contributed to increase the performance of the algorithms?

Additional comments

This study reports results on the performance of random forest algorithms for the recognition of daily life activities using plantar pressure information collected with a smart-shoe. For these goals, authors enrolled 17 female healthy subject with mean age of 26 ± 9 years old who took par in the experimental study. Their results confirmed the possibility of high-performance human behavior recognition using plantar pressure data only. The study is interesting. The manuscript is well written and conclusions are supported by reported results. There are only few aspects I would like authors to comment and possibly consider the above questions to improve the quality of their manuscript.

---

## Round 0.2 · Minor Revisions

The authors have done a fine job in addressing many of the comments of the two reviewers. However, reviewer one still has some small recommendations for you to consider prior to this manuscript being acceptable for publication in PeerJ.

Reviewer 1 ·

Basic reporting

NA

Experimental design

1) In Figure 4, the color bar in stage 2 is unclear. What does "expected to perform" mean? It should belong to "RESULTS"

Validity of the findings

NA

---

## Round 0.3 · accepted · Accept

I would like to thank the authors for their attention to addressing all of the comments provided by the reviewers and me. I would therefore like to recommend this paper be accepted for publication in PeerJ.